# Widespread inter-individual gene expression variability in *Arabidopsis thaliana*

Sandra Cortijo, Zeynep Aydin, Sebastian Ahnert & James CW Locke*

## Abstract

A fundamental question in biology is how gene expression is regulated to give rise to a phenotype. However, transcriptional variability is rarely considered although it could influence the relationship between genotype and phenotype. It is known in unicellular organisms that gene expression is often noisy rather than uniform, and this has been proposed to be beneficial when environmental conditions are unpredictable. However, little is known about inter-individual transcriptional variability in multicellular organisms. Using transcriptomic approaches, we analysed gene expression variability between individual *Arabidopsis thaliana* plants growing in identical conditions over a 24-h time course. We identified hundreds of genes that exhibit high inter-individual variability and found that many are involved in environmental responses, with different classes of genes variable between the day and night. We also identified factors that might facilitate gene expression variability, such as gene length, the number of transcription factors regulating the genes and the chromatin environment. These results shed new light on the impact of transcriptional variability in gene expression regulation in plants.

**Keywords** *Arabidopsis thaliana*; inter-individual heterogeneity; noise in gene expression; RNA-seq; transcriptional variability
**Subject Categories** Genome-Scale & Integrative Biology; Plant Biology; Transcription
**Mol Syst Biol. (2019) 15: e8591**

## Introduction

Gene expression in individual cells is often noisy and dynamic. Genetically identical cells under the same environment can display widely different expression levels of key genes (Ko, 1992; Fiering *et al*, 2000; Martins & Locke, 2015). Noise in gene expression has been shown to have a significant impact on the design and function of genetic circuits in unicellular organisms (Elowitz *et al*, 2002; Eldar & Elowitz, 2010). It has also been observed in multiple pathways in mammalian cells (Yin *et al*, 2009; Mantsoki *et al*, 2016; Riddle *et al*, 2018), in *Drosophila* cells (Pare *et al*,

2009) and between individuals in *Drosophila* (Lin *et al*, 2016). However, gene expression variability has mostly been analysed for a few individual genes in plants at a single-cell resolution (Angel *et al*, 2015; Araujo *et al*, 2017; Meyer *et al*, 2017; Gould *et al*, 2018). Several studies suggest that transcriptional variability between cells can be exploited during development in multiple organisms (Wernet *et al*, 2006; Chang *et al*, 2008; Pare *et al*, 2009; Meyer *et al*, 2017). On the other hand, the identification of mutants in which transcriptional and/or phenotypic variability is increased indicates that variability is at least partly buffered or controlled (Rutherford & Lindquist, 1998; Queitsch *et al*, 2002; Raj *et al*, 2010; Folta *et al*, 2014; Schaefer *et al*, 2017). It is not known at a genome-wide scale to what extent gene expression can be variable during plant development or between identical plants.

Plants are a promising system to examine the global properties of noise in gene expression, as phenotypic variability, also referred to as phenotypic instability, has been observed in multiple areas of plant growth and development. Inter-individual phenotypic variability has also been observed in isogenic populations of other organisms (Zhang *et al*, 2016; Roman *et al*, 2018), but the model plant *Arabidopsis thaliana* presents the advantage of being an inbreeding species where heterozygosity is extremely low (Abbott & Gomes, 1989). Phenotypic variability in plants can occur both within and between individuals that are growing in the same conditions. High levels of phenotypic variability have been described for seed germination time (Simons & Johnston, 2006; Venable, 2007; Mitchell *et al*, 2017), patterning of lateral roots (Forde, 2009) as well as for floral and foliar development (Paxman, 1956; Sakai & Shimamoto, 1965). It is not known whether such inter-individual phenotypic variability originates from responses to microenvironmental perturbations, or from stochastic factors at the cellular level or from both. Differences in the level of inter-individual variability have been observed between natural accessions, in recombinant inbred lines and also in mutants for many traits such as growth, hypocotyl length, leaf and flower number, plant height and plant defence metabolism (Hall *et al*, 2007; Jimenez-Gomez *et al*, 2011; Folta *et al*, 2014; Hong *et al*, 2016; Schaefer *et al*, 2017). Jimenez-Gomez and colleagues also identified QTLs explaining differences in the level of expression variability between pools of plants in *Arabidopsis thaliana* (Jimenez-Gomez *et al*, 2011). This suggests that such variability can be controlled or buffered by genetic factors.

The Sainsbury Laboratory, University of Cambridge, Cambridge, UK
*Corresponding author. Tel: +44 1223 761110; E-mail: james.locke@slcu.cam.ac.uk

However, the molecular mechanisms underlying such inter-individual phenotypic variability are still poorly understood.

In this work, we analyse gene expression variability between multicellular individuals using the model plant *Arabidopsis thaliana*, with the emphasis on three questions. Firstly, what is the global extent of gene expression variability between individuals? In order to better understand how gene expression variability is controlled and its role in plant physiological and developmental responses, we first need to identify genome-wide the genes that are highly variable between individuals. Secondly, does inter-individual expression variability change through the diurnal cycle? It is known that the diurnal cycle influences expression level of up to 36% of the transcriptome (Michael & McClung, 2003; Covington *et al*, 2008). However, little is known about the impact of the diurnal cycle on gene expression variability. Thirdly, what factors can regulate this inter-individual expression variability?

Using single seedling RNA-seq, we identified hundreds of genes that are highly variable between individuals in *Arabidopsis* and show that the level of variability changes throughout the diurnal cycle. To ensure accessibility and reusability of our data, we created an interactive web application, in which the inter-individual gene expression variability through a diurnal cycle can be observed for individual genes (https://jlgroup.shinyapps.io/aranoisy/). This tool will help researchers to take into consideration any inter-individual gene expression variability in their genes of interest. Moreover, we show that highly variable genes (HVGs) are enriched for environmentally responsive genes and characterised by a combination of specific genomic and epigenomic features. We have revealed both the level and potential mechanism behind gene expression variability between individuals in *Arabidopsis*, allowing understanding of a previously unexplored aspect of gene regulation during plant development.

# Results

### Widespread expression variability in *Arabidopsis* seedlings through the day and night

In order to measure transcriptional variability between individuals, we generated transcriptomes for single *Arabidopsis thaliana* seedlings at multiple time-points over a full day/night cycle (Fig 1A). To minimise any variability caused by external factors, these seedlings originated from the same mother plant, germinated at the same time and were grown in the same plate under controlled conditions (see Materials and Methods for more details). To analyse how transcriptional variability is influenced by diurnal cycles, we harvested seedlings every 2 h across a 24-h period (Fig 1A). ZT2 to ZT12 corresponding to the time-points harvested during the day, and ZT14 to ZT24 to the time-points harvested during the night, ZT12 and ZT24 being, respectively, harvested just a few minutes before dusk and dawn. In total, 168 transcriptomes have been analysed, that is, of 14 individual seedlings for each of the 12 time-points. We observed very similar mean expression profiles to already published bulk level diurnal profiles (Mockler *et al*, 2007; Appendix Fig S1B), indicating that known diurnal expression patterns are reproduced in our experiment.

Before identifying highly variable genes, we tested whether the level of variability observed in our data could be explained by experimental error or be due to the RNA-seq method. In order to validate the profiles for gene expression variability during the time course, we performed a full time course replicate and examined the variability between seedlings for 10 genes by RT–qPCR (Appendix Fig S1C and G). We observed very similar expression profiles for these genes (Appendix Fig S1G). We also found a positive correlation of 0.77 (Spearman correlation, $P = 0.0092$) between the average $CV^2$ of the entire time course for each of these 10 genes measured by RNA-seq or RT–qPCR (Appendix Fig S1C), indicating that genes have a similar level of variability in both experiments. We also obtained very similar $CV^2$ at ZT24 when using independent mapping programs (salmon compared to TopHat with a Spearman correlation of 0.85, $P < 2.2e-16$, or hisat2 compared to TopHat with a Spearman correlation of 0.9, $P < 2.2e-16$, Appendix Fig S1E and F).

Having validated the measured inter-individual variability, we identified highly variable genes (HVGs) from our RNA-seq data set using a previously described method (Brennecke *et al*, 2013). In this method, the square coefficient of variation [$CV^2$ = variance/(average$^2$)] of each gene is compared to its average expression level (Fig 1B). In order to avoid biases caused by technical noise, which is likely to be higher at lower expression levels, we only selected the HVGs if they were significantly more variable than the background trend in $CV^2$ (see Materials and Methods for more detail). We also calculated a corrected $CV^2$ for each gene [$\log_2(CV^2/trend)$] which corrected for the observed negative trend between $CV^2$ and expression level, and used it for further analyses of gene expression variability. Genes with a negative $\log_2(CV^2/trend)$ are less variable than the trend, while genes with a positive $\log_2(CV^2/trend)$ are more variable than the trend. To test whether there were transcriptome-wide trends in the level of variability across the day, we verified that the global trends of the $CV^2$ against the average normalised expression measured for each time-point are in the same range (Appendix Fig S2A). We also observed that there was no obvious bias in the distribution of $CV^2$ against the average normalised expression at the different time-points (Appendix Fig S2B). We observed that while some genes are never classed as a HVG (Fig 1C left and Appendix Fig S2D), others are selected as a HVG for the entire time course (Fig 1C middle and Appendix Fig S2E), or as a HVG for only some time-points (Fig 1C right and Appendix Fig S2F), indicating a broad range of variability profiles during the diurnal cycle. Expression level in individual seedlings and profiles of the $\log_2(CV^2/trend)$ during the time course for individual genes can be viewed at https://jlgroup.shinyapps.io/AraNoisy/ (see Appendix Fig S2H for more detail about how to use the web application).

In total, we identified between 257 and 716 HVGs at each time-point, with more HVGs identified during the night (Fig 2A). We also generated two other reference gene lists to compare to the HVGs: 1,000 randomly selected sets of genes for each time-point, with the number of random genes selected matching the number of HVGs for each time-point (Appendix Fig S3B), and the least 1,000 variable genes for each time-point (LVGs, for lowly variable genes, see Appendix Fig S3A). We see that HVGs are at least three times and on average 9.3 times more variable than the global trend, while LVGs are at least 4.8 times and on average 8.9 times less variable than the global trend (Appendix Fig S2C). Random genes span a wide range of variability including values as low as for LVGs and as

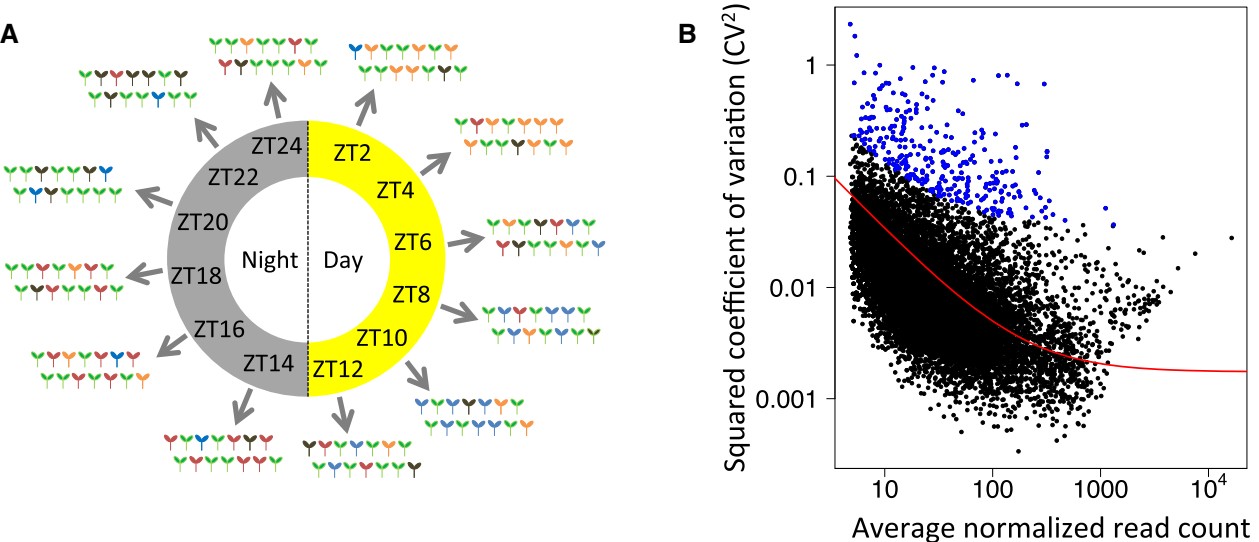

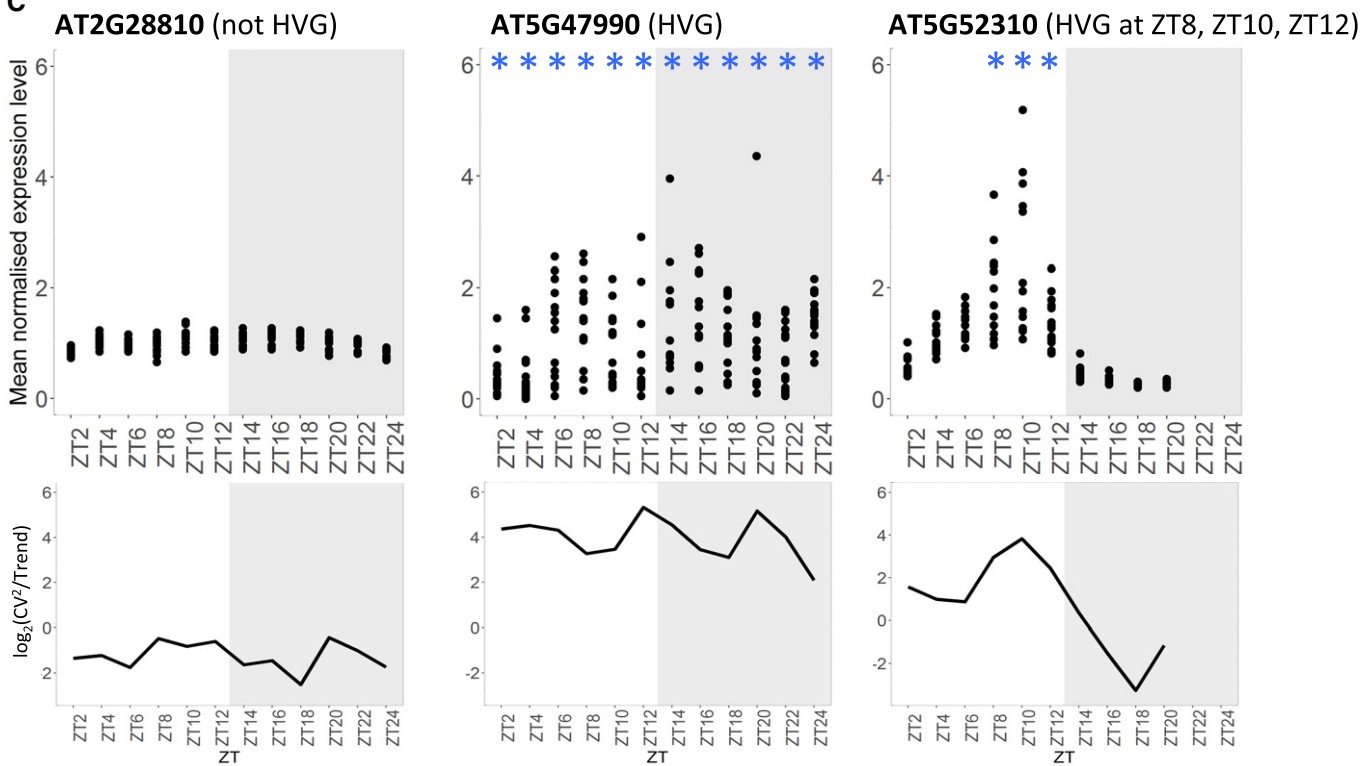

**Figure 1.   Widespread expression variability in *Arabidopsis seedlings* through the day and night.**

A   Experimental set-up to identify transcriptional variability between seedlings during the day and night. RNA-seq was performed on individual seedlings, for a total of 14 seedlings at each time-point. Seedlings were harvested at 12 time-points, every 2 h across a 24-h period. Seedlings of different colours represent different transcriptional states and thus inter-individual expression variability we aim to identify.

B   Identification of highly variable genes (HVGs) for the time-point ZT2. The red line shows the trend for the global relation between $CV^2$ (variance/mean$^2$) and mean expression, which is defined using all genes (minus small and lowly expressed genes, see Materials and Methods for more detail) and used to identify HVGs (blue points). For each gene, a corrected $CV^2$ is calculated: $\log_2(CV^2/\text{trend})$.

C   Expression profiles (top) in the 14 seedlings over a 24-h time course (with 12 time-points) for a non-variable gene (left), a highly variable gene (middle) and a gene with the level of variability changing across the 24 h (right). Each dot is the mean normalised expression level for a single seedling. Variability profiles (bottom) of the $\log_2(CV^2/\text{trend})$ for the same genes are also shown. Blue stars indicate time-points for which the gene is identified as being highly variable.

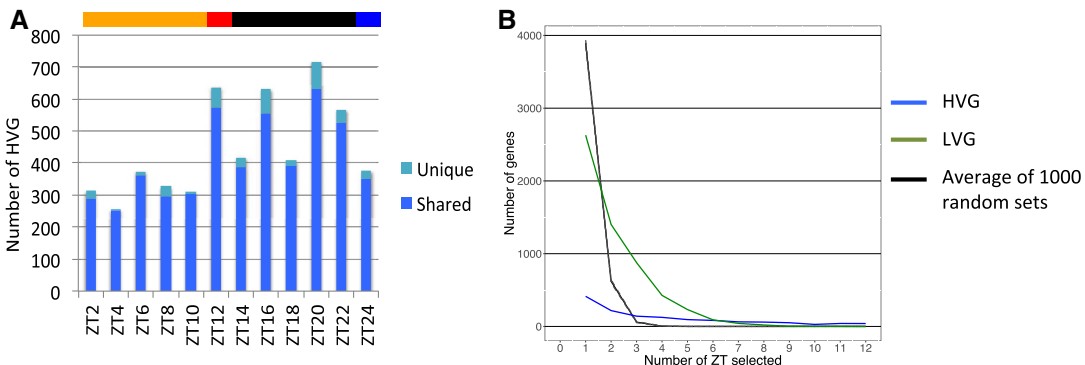

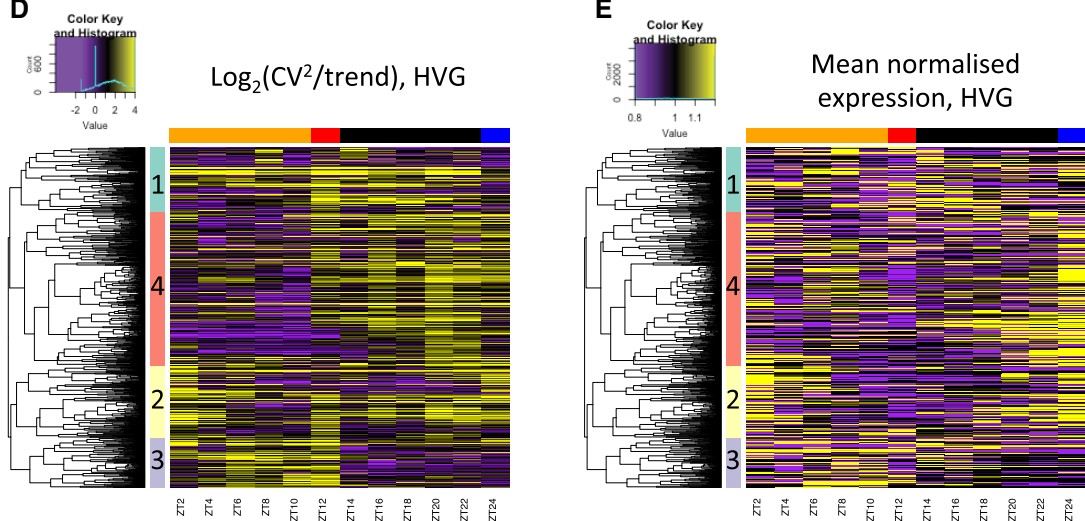

**Figure 2.   Structure of noisiness shows partitioning between day and night.**

A  Number of genes identified as being highly variable for each time-point. These genes are separated between those that are also selected as highly variable in at least one other time-point (dark blue) and those highly variable in only one time-point (light blue). The top bar indicates time-points harvested during the day (orange), just before dusk (red), during the night (black) and just before dawn (blue).

B  Distribution of the number of time-points at which genes are identified as highly variable (blue) or lowly variable (green). The distribution of average (black) and 95% confidence interval (dotted grey) for the thousand random sets are also represented and are so close that they are superimposed and cannot be differentiated in the figure.

C  Heatmap of the percentage of HVGs shared between time-points. Red indicates a high percentage of HVGs in common between two time-points. The top and side bars indicate time-points harvested during the day (orange), just before dusk (red), during the night (black) and just before dawn (blue).

D  Hierarchical clustering of HVGs based on the $\log_2(CV^2/\text{trend})$ at each time-point. The result is represented as a heatmap where yellow indicates a high $\log_2(CV^2/\text{trend})$. The genes were separated into four clusters, indicated by the side coloured bar. The top bar indicates time-points harvested during the day (orange), just before dusk (red), during the night (black) and just before dawn (blue). See Appendix Fig S3G for heatmaps of the $\log_2(CV^2/\text{trend})$ with the same colour cut-offs for HVGs, LVGs and random genes.

E  Heatmap of the mean normalised expression level for the genes in Fig 2D, keeping the same clustering organisation. The top bar indicates time-points harvested during the day (orange), just before dusk (red), during the night (black) and just before dawn (blue).

high as for HVGs. All these results taken together reveal a widespread level of inter-individual variability in gene expression throughout the entire time course, with 8.7% of the analysed transcriptome identified as highly variable in at least one time-point (1,358 HVGs). Moreover, as we describe in more detail below, some genes can show very different levels of variability during the time course.

## Structure of noisy genes shows partitioning between day and night

We next examined the structure of our measured variability in more detail. First, we examined whether HVGs identified for each time-point were scored as variable for multiple time-points or for this time-point only. The vast majority (~93%) of the HVGs identified for each time-point are also identified as highly variable in another time-point (Fig 2A, dark blue). In comparison, an average of ~80% of LVGs and ~30% of random genes selected for each time-point is also observed in another time-point (Appendix Fig S3A and B).

Since many genes are identified in more than one time-point, we then defined in how many time-points they are identified. We performed this analysis on all the HVGs (1,358 HVGs), LVGs (5,727) and on a thousand sets of random genes selected for each time-point (same number as HVGs for the time-point). In total, 30% of all the 1,358 HVGs are identified in only one time-point while the others are shared with other time-points, up to all of the 12 time-points for 40 genes (Fig 2B). LVGs and random genes are identified in a lower number of time-points, with 46% of all LVGs and on average 85% of all random genes that are specific for one time-point, while no genes are observed as lowly variable or random in all 12 time-points (Fig 2B). These results show that the number of HVGs shared between time-points is higher than what is observed for LVGs and random genes. It indicates that genes can be highly variable for multiple time-points and potentially show profiles in gene expression variability.

Having observed that HVGs are shared between time-points, we then tested for any structure in the gene expression variability across the diurnal cycle. We calculated the percentage of HVGs that are shared between any two time-points and can see a clear separation of day and night time-points (Fig 2C). ZT12, which was harvested just a few minutes before dusk, behaves more like a night than a day time-point as it shares a higher proportion of HVGs with night time-points. When excluding ZT12, the percentage of HVGs that are shared between two time-points of the day (~55% on average) and two time-points of the night (~60%) is higher than between one time-point of the day and one time-point of the night (~35%). When doing the same analysis for LVGs, we observed that the percentage of genes that are shared between two time-points of the day (~18.5%) and two time-points of the night (~20.8%) is very similar to the percentage of genes shared between one time-point of the day and one time-point of the night (~17%, Appendix Fig S3C). We could not find any difference in the percentage of genes that are shared between two time-points in these three categories for the set of random genes analysed (Appendix Fig S3D). This result indicates a structure of the HVGs, but not of LVGs and random genes, in the time course, with a separation between day and night.

In order to identify profiles of inter-individual variability across the time course, we performed hierarchical clustering of all 1,358 HVGs based on their $\log_2(CV^2/\text{trend})$ at each time-point. We identified four clusters of variability patterns across the time course (Fig 2D, Appendix Fig S3G). Profiles of genes representative of each cluster can be seen in Appendix Fig S3H. Two clusters (543 genes, clusters 1 and 2) are composed of genes that are variable during the day and the night. One cluster (200 genes, cluster 3) is composed of genes that are highly variable mainly during the day, while another one (615 genes, cluster 4) is composed of genes highly variable mainly during the night. This observation is specific for HVGs, as we cannot observe such marked structure of variability profiles for LVGs and a set of random genes (Appendix Fig S3E–G). All these results show a clear structure in gene expression variability between seedlings during the time course, with different sets of genes being variable during the day or during the night, further suggesting regulation of the level of variability.

## Highly variable genes are enriched in environmentally responsive genes

We next examined the function of the HVGs and LVGs. To do so, we first analysed Gene Ontology (GO) enrichment for all 1,358 HVGs. We identified enrichment for several processes involved in the response to biotic and abiotic stresses as well as in the response to endogenous and exogenous signals (Table EV3). This is not the case for the 5,727 LVGs, for which we found enriched GOs involved in primary metabolism (Table EV4), or for the random genes, for which no GO term was enriched.

Interestingly, different GOs are enriched in the clusters identified in Fig 2D based on the $\log_2(CV^2/\text{trend})$ of HVGs along the time course (Table EV3). For example, the response to cold is enriched in clusters 1 and 3, containing genes highly variable during the day, while nitrate assimilation is only enriched in cluster 4, which contains genes highly variable during the night (Table EV3). This result suggests that some GOs might be variable at specific times of the diurnal cycle. In order to test this, we analysed GO enrichment for the HVGs identified at each time-point and clustered the GOs based on the $\log_{10}(\text{FDR})$ of their enrichment at the different time-points (Fig 3A). While some GOs such as lipid transport and defence response to fungus are enriched in HVGs throughout the entire time course, we also identified GOs that are enriched only for a subset of the time course. This is the case for the response to toxic substance, reactive oxygen species metabolic process and response to iron ion that are more enriched during the night, or the response to water deprivation and to cold that are more enriched during the day (Fig 3A). We also analysed GO enrichment for the LVGs at each time-point and do not observe such enrichment of GOs preferentially during the day or night (Fig 3B, Table EV4). We also observed that HVGs tend to be expressed with a higher tissue specificity compared with LVGs and random genes (Appendix Fig S4A, Wilcoxon text $P < 2.2e^{-16}$ between HVG and LVG, Wilcoxon text $P < 2.2e^{-16}$ between HVG and the thousand sets of random genes). Most HVGs are still expressed in more than one tissue (Appendix Fig S4B). This higher tissue specificity is in agreement with the enrichment of many GOs associated with tissue-specific functions in HVGs.

In order to support these GO enrichment results, we also examined the transcription factors binding to the HVGs and LVGs, using available data generated by DNA affinity purification coupled with

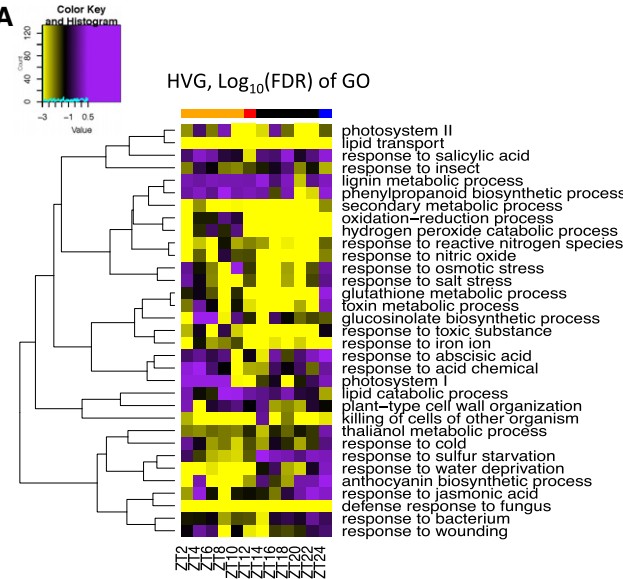

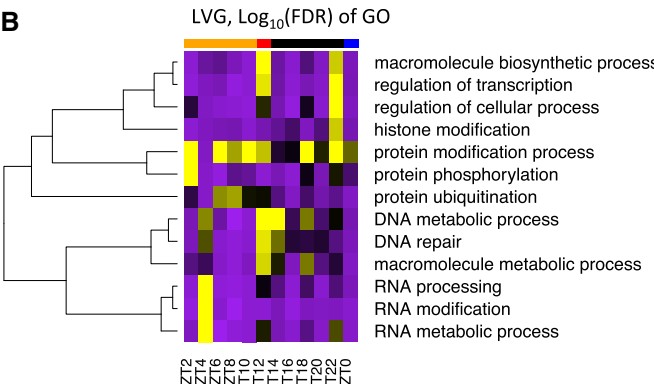

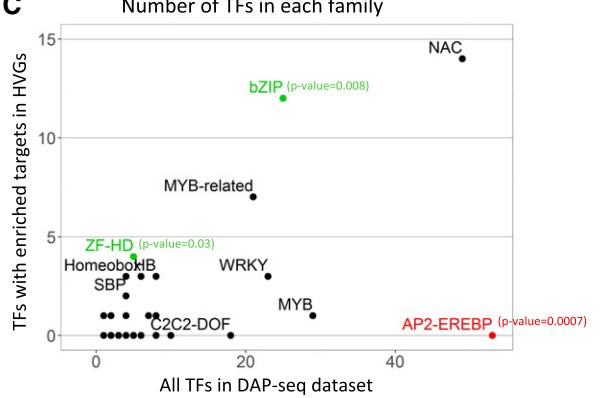

**Figure 3.  HVGs are enriched for stress responses.**

A   GO enrichment for genes selected as highly variable for each time-point. GOs that are enriched in at least one time-point are represented. Hierarchical clustering of the GO is performed on the $\log_{10}$(FDR) for GO enrichment in the HVGs. The result is presented as a heatmap with significantly enriched GO in yellow. The top bar indicates time-points harvested during the day (orange), just before dusk (red), during the night (black) and just before dawn (blue).

B   GO enrichment for genes selected as lowly variable for each time-point. GOs that are enriched in at least one time-point are represented. Hierarchical clustering of the GO is performed on the $\log_{10}$(FDR) for GO enrichment in the LVGs. The result is presented as a heatmap with significantly enriched GO in yellow. The top bar indicates time-points harvested during the day (orange), just before dusk (red), during the night (black) and just before dawn (blue).

C   Number of transcription factors (TF) in each TF family with enriched targets in the HVGs compared with the total number of TFs in each family included in the DAP-seq data. Families in green have a significantly higher number of TFs with enriched targets in the HVGs than in the entire data set. Families in red have a significantly lower number of TFs with enriched targets in the HVGs than in the entire data set (based on a Fisher's exact test for which *P*-values are included in figure).

derived from *in vitro* interaction, it only provides a list of potential targets and further experiments such as ChIP-seq would be required to obtain the list of genes regulated by these TFs in our conditions. When deriving gene regulatory networks from the DAP-seq data for HVGs and these TFs, we observed a high level of regulation of these 60 TFs by other TFs of this same list and that most HVGs are targeted by a combination of highly variable and non-highly variable TFs (Appendix Fig S4C–E and Table EV6). These results suggest that while the high level of variability could potentially partly be explained by TFs, other factors are also probably involved. These 60 TFs with enriched targets in the HVGs are mainly part of the NAC-, bZIP- and MYB-related families. ZF-HD (4 TFs with enriched targets in the HVGs) and bZIP (12 TFs with enriched targets in the HVGs) families are significantly more represented in this set than expected based on the entire DAP-seq data set (Fig 3C). bZIP TFs regulate multiple processes including pathogen defence, light and stress signalling, seed maturation and flower development (Jakoby *et al*, 2002). These are in agreement with the enriched GOs identified for HVGs, involved in responses to the environment as well as biotic and abiotic stresses.

These results show that HVGs are enriched for genes involved in the response to environment and stress and are targeted by TF families involved in environmental responses, while LVGs are enriched in DNA, RNA and protein metabolism. Moreover, the clear pattern of enrichment for some function either during the day or the night further suggests that variability between seedlings across the day and night is functional and might be controlled.

## Gene expression profiles and variability are not correlated for the majority of the HVGs

Having identified that HVGs are enriched in stress-responsive genes and that variability is structured during a diurnal cycle, we next asked what factors could be involved in modulating variability in expression? To test whether expression levels could modulate variability, we analysed the expression level of HVGs, LVGs and random genes at each time-point (Appendix Fig S5A). We observed

sequencing (DAP-seq), which provides the list of *in vitro* targets for 529 TFs (O'Malley *et al*, 2016). We identified 60 TFs with enriched targets in the HVGs, 5 TFs with enriched targets in the LVGs and only one TF with enriched targets in the random genes. Out of the 60 TFs with enriched targets in the HVGs, only 7 are themselves HVGs. 1,106 out of 1,358 HVGs are potential targets of at least one of these 7 TFs. However, 23,301 genes in total are potential targets of at least one of these 7 TFs, so only a small fraction of these potential targets are HVGs (Table EV5). Moreover, DAP-seq data being

that expression levels of HVGs are slightly higher than for a thousand sets of random genes and LVGs. In order to define whether expression profiles could influence changes in variability during the time course, we used the same hierarchical clustering of HVGs based on their $\log_2(CV^2/\text{trend})$ (Fig 2D) to represent their mean normalised expression levels (Fig 2E). We observed that genes in cluster 4, which are more variable during the night, have a peak of expression at the end of the night. Genes in cluster 3, that are more variable during the day, are also slightly more expressed during the day. However, at a global scale, we cannot see a general link between profiles of gene expression level and variability for all HVGs (Fig 2E).

To go further, we analysed the correlation between the profiles of mean normalised expression and of the $\log_2(CV^2/\text{trend})$ profile for each of the 1,358 HVGs, 5,727 LVGs as well as for a thousand sets of random genes of the same number as all HVGs (1,358, Appendix Fig S5B). We observed a slightly higher correlation for HVGs (median of 0.18) compared to LVGs (median of −0.06) and the thousand sets of 1,358 random genes (median of −0.005 on average with a 95% confidence interval of −0.019 to 0.008). However, while variability and expression levels are positively correlated for some HVGs (peak around 0.5 in Appendix Fig S5B), this is not the case for many other HVGs (peak around 0 in Appendix Fig S5B). We cannot see major differences between these two groups of genes in the expression profiles or the number of time-points for which genes are identified as HVGs (Appendix Fig S5C and D). However, HVGs with a positive correlation between variability and expression levels (peak around 0.5 in Appendix Fig S5B) have a lower expression level in general compared to other HVGs (Appendix Fig S5E). If we consider HVGs with a significant correlation ($P \leq 0.05$), it seems that profiles in gene expression variability for approximately 20% of HVGs could be potentially explained by expression profiles (for profiles of positive and negative correlations, see examples in Appendix Fig S5F and G). No significant correlation can be measured between gene expression and variability profiles for the remaining 80% of the genes (example Appendix Fig S5H). Altogether, these results suggest that profiles in variability could potentially be explained by expression levels for only a fifth of HVGs, indicating that other factors might be involved in facilitating gene expression variability.

## Noisy genes tend to be smaller and to be targeted by more transcription factors

In order to identify other factors that might be involved in regulating gene expression variability, we analysed several genomic features including gene length, number of introns and the number of TFs targeting the genes for all 1,358 HVGs, 5,727 LVGs and a thousand sets of 1,358 random genes. We first observed that HVGs tend to be shorter and contain a lower number of introns than LVGs or random genes (Fig 4A and B, Appendix Fig S6A and B, Wilcoxon text $P < 2.2e^{-16}$ between HVG and LVG, Wilcoxon text $P < 2.2e^{-16}$ between HVG and the thousand sets of random genes). We also observed a negative trend between the level of variability and the gene length or number of introns for all genes at each time-point (Appendix Fig S6C and D). As the gene length and number of introns are strongly positively correlated (Appendix Fig S6E), we analysed the impact of one of these factors on gene expression

variability while fixing the other and vice versa. We observed very similar distributions for the number of introns of HVGs, LVGs and thousand sets of random genes when these genes are of similar size (Appendix Fig S6F). On the contrary, we observed a trend for HVGs to be smaller when comparing genes with the same number of introns, for genes with three introns and less (Appendix Fig S6G). These results suggest that gene length might have a more important role than the number of introns in facilitating gene expression variability. In order to check for potential experimental bias that could account for the fact that smaller genes are more variable, we fragmented *in silico* 27 genes of ~1.5 to ~2.5 kb into smaller fragments of ~250–300 bp and examined whether this could affect the level of gene expression variability that we estimate (Appendix Fig S6H). We performed this analysis for genes with different levels of expression that are either HVGs, LVGs or have a corrected $CV^2$ around zero (i.e. close to the global trend). We observed a very similar level of corrected $CV^2$ for full genes and their fragments (Appendix Fig S6H). Only 2 fragments out of the 35 (5%) originating from HVGs are not any more identified as highly variable, and only 2 fragments out of the 63 (3%) originating from genes that are not highly variable are now identified as highly variable. These results suggest that the trend we observe of HVGs to be smaller is not caused by technical biases.

One other factor we tested is the binding of transcription factors (TFs) at the promoters of genes. For this, we counted the number of TFs binding to the promoter for all 1,358 HVGs, 5,727 LVGs and the thousand sets of 1,358 random genes using the available DAP-seq data and found a tendency for a higher number of TFs binding the promoter of HVGs (Fig 4C, Wilcoxon text $P < 2.2e^{-16}$ between HVG and LVG, Wilcoxon text $P$-value of 0.079 to 7.2e$^{-09}$ between HVG and the thousand sets of random genes). This result suggests differences in the way HVG and LVG expression are regulated, which could possibly be due to different network architectures.

## Noisy genes tend to have a chromatin environment refractory to expression

On top of genomic features, another factor that can influence gene expression is the chromatin structure. In order to identify whether HVGs are characterised by a specific chromatin structure, we analysed several histone marks using data already available (for which we have no information about the time of day when the plants were harvested). We first analysed the proportion of genes containing a histone modification among all 1,358 HVGs, 5,727 LVGs and the thousand sets of random genes in comparison with all background genes. We could identify that HVGs are enriched in H3K27me1 and H3K27me3, which are repressive marks, while they are depleted in active marks such as H3K4me2, H3K4me3, H3K36me3 or H2Bub (Fig 5A). They are also depleted in DNA methylation (Fig 5A), which is usually considered as a permissive mark for expression when in the body of genes (Zilberman *et al*, 2007; Coleman-Derr & Zilberman, 2012). On the other hand, LVGs are enriched in these active marks and depleted in H3K27me1 and H3K27me3. From previous studies, genes containing H2A.Z histone variant have been separated into two classes: (i) genes with a high signal in the gene body, which are enriched for environmentally responsive genes and genes with tissue-specificity expression, and (ii) genes with a low signal in the gene body for which H2A.Z is mainly observed at the

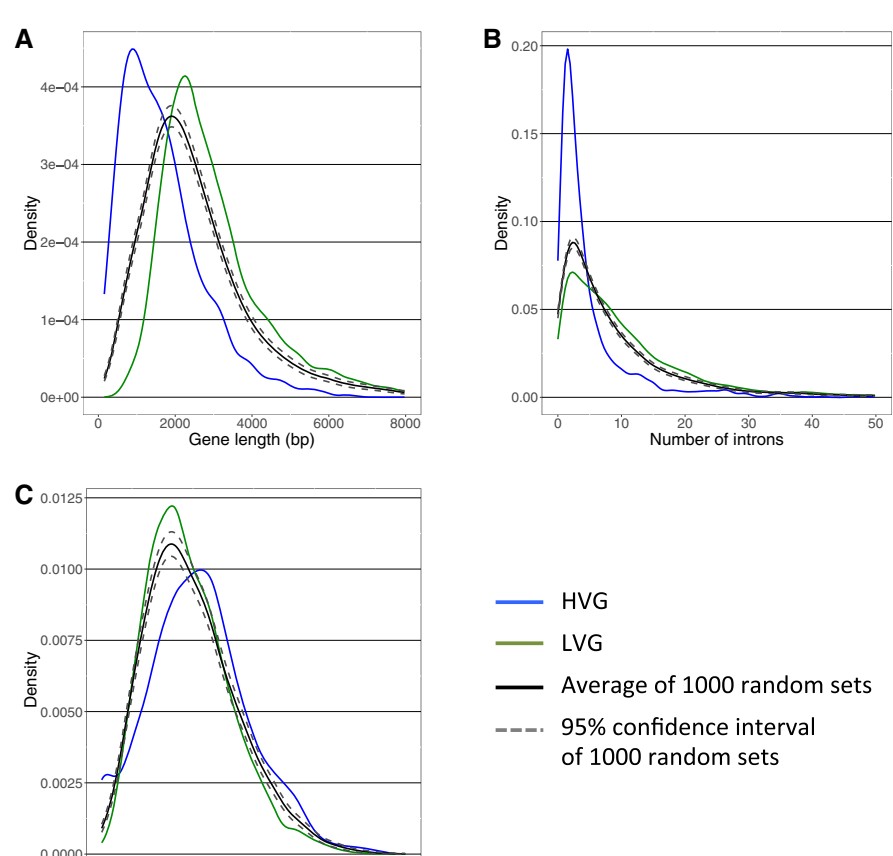

**Figure 4.   HVGs tend to be small and to be targeted by a higher number of TFs.**

A   Distribution of the gene length (in bp) for LVGs (green) and HVGs (blue). The distribution of average (black) and 95% confidence interval (dotted grey) for the thousand random sets is also represented.

B   Distribution of the number of introns for LVG (green) and HVG (blue). The distribution of average (black) and 95% confidence interval (dotted grey) for the thousand random sets is also represented.

C   Distribution of the number of TFs targeting a gene, based on the DAP-seq available data set, for LVG (green) and HVG (blue). The distribution of average (black) and 95% confidence interval (dotted grey) for the thousand random sets is also represented.

1st nucleosome, which are enriched for housekeeping genes (Coleman-Derr & Zilberman, 2012). The former category is enriched among HVGs, while the latter category is enriched among LVGs (Fig 5A).

To define whether HVGs and LVGs are also characterised by different profiles for these chromatin marks, rather than just differing in their presence/absence, we used already published ChIP-seq data for several chromatin marks (for which we have no information about the time of day when the plants were harvested) and represented the signal along the genes. We identified differences in the profiles of the average chromatin signal between HVGs, LVGs and random genes for H3K27me3, H2A.Z, H3K4me3 and H3K23ac (Fig 5B). A higher H3K27me3 average signal is observed for HVGs (Fig 5B) and can be explained by a higher number of HVGs containing this mark compare to LVGs and random genes (Fig 5C). We observed H2A.Z and H3K23ac signal throughout the gene body for HVGs, while LVGs and random genes are characterised by a peak around the TSS, corresponding to the 1st nucleosome, and a lower signal for the rest of the gene body (Fig 5B and C). We see a higher H3K4me3 average

signal for LVGs and random genes characterised with a peak at the beginning of the genes, while less than half of the HVGs have a high signal for this chromatin mark. To correct for differences in gene size between HVGs and LVGs, we also performed the same analysis on a subset of 150 HVGs and 185 LVGs and 125 random genes that have a similar size of 1,100–1,400 bp (Appendix Fig S7A). The results are broadly the same as the ones obtained on all HVGs and LVGs. These results indicate that HVGs are characterised by a more compacted chromatin environment, as further supported by the fact that the MNase signal, which indicates the level of nucleosome occupancy, is higher in the gene body and mostly at the end of the genes in HVGs compared to LVGs and random genes (Appendix Fig S7B).

In summary, HVGs and LVGs are characterised by a specific chromatin environment, in terms of the presence/absence of chromatin marks as well as for the profiles of these marks. Our results indicate that chromatin at HVGs tends to be more compacted and refractory to expression than at LVGs and random genes, which might have implications for how expression is regulated in these genes.

                                   

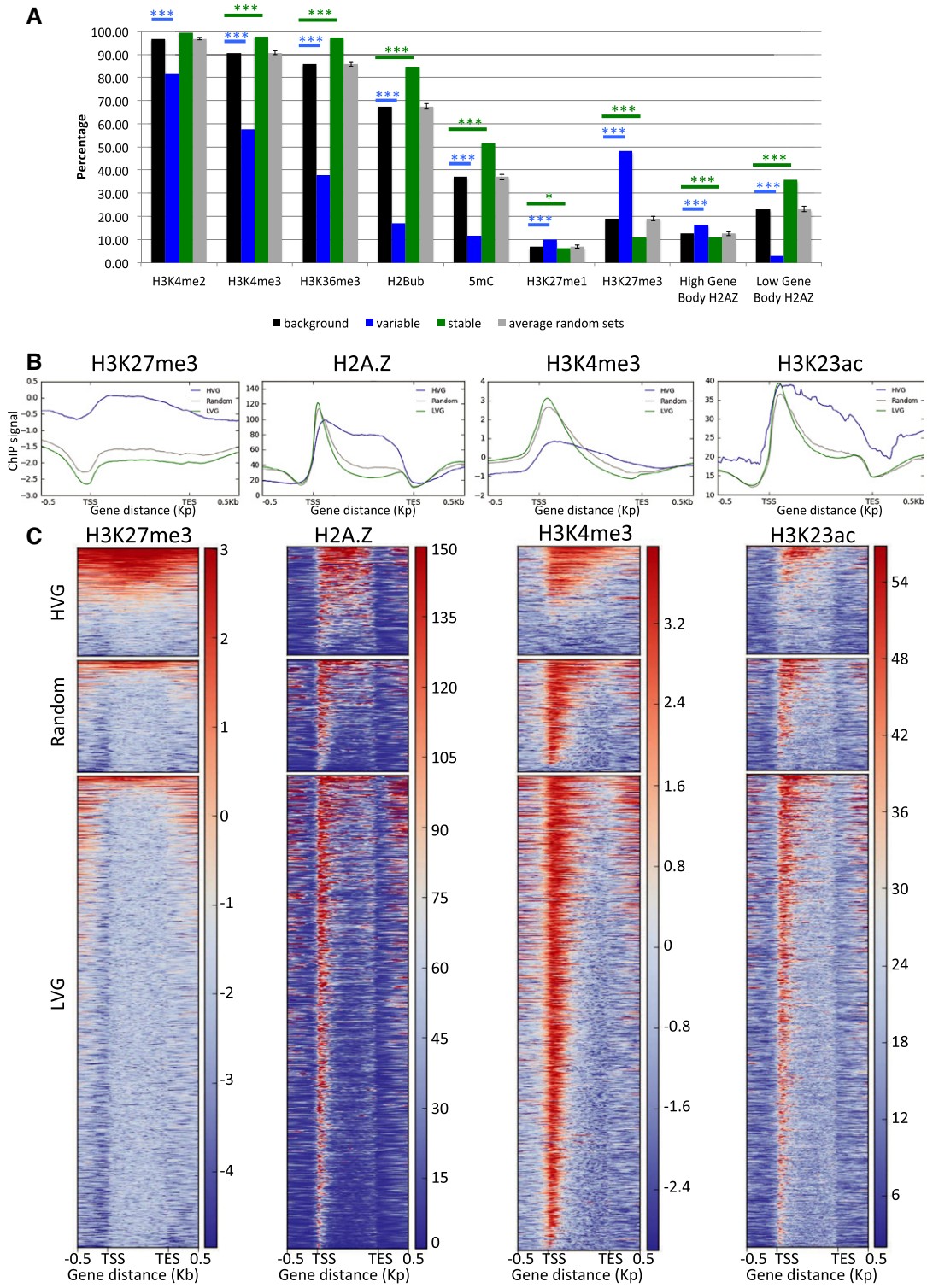

**Figure 5. HVGs tend to have a specific chromatin environment.**

A  Proportion of genes marked with several chromatin marks among all genes passing size and expression level thresholds (black), HVGs (blue), LVGs (green) and the average for the thousand random sets (grey). Error bars indicate the 95% confidence interval for the thousand random sets. Blue and green stars indicate statistical differences in the proportion of marked genes compared to all genes, for variable and stable genes, respectively (* indicates a $P < 0.05$, *** indicates a $P < 0.001$, chi-square test).

B  Average profile for H3K27me3, H2A.Z, H3K4me3 and H3K23ac at HVGs (blue), LVGs (green) and random genes (grey).

C  Heatmap of the enrichment for H3K27me3, H2A.Z, H3K4me3 and H3K23ac for HVGs (top), random genes (middle) and LVG (bottom). Red means a high level and blue means a low level for the chromatin marks.

# Discussion

In this work, we have characterised the variability in gene expression between individual *Arabidopsis* seedlings at the genome-wide scale throughout a diurnal cycle. To do this, we have analysed 14 seedlings at each of the 12 time-points, generating 168 transcriptomes in total. This resource reveals previously unexplored variability for multiple pathways of interest for plant researchers, as well as providing insights into the modulation of gene expression variability at the genome-wide scale (Fig 6). We have successfully identified highly variable genes across the diurnal cycle, finding two sets of genes variable either during the day or night (Fig 2), revealed the functional classes of highly variable genes (Fig 3) as well as their genomic and epigenomic characteristics (Figs 4 and 5). Interestingly, most profiles of gene expression variability are not correlated with profiles in gene expression levels (Appendix Fig S5), indicating that changes in expression levels during the diurnal cycle are not sufficient to explain changes in inter-individual variability. The large degree of gene expression variability revealed by our study will impact on our functional understanding of pathways as well as experimental design. To enable researchers to access this resource, we have created a graphical web interface to allow easy visualisation of inter-individual gene expression variability during a diurnal cycle for genes of interest (https://jlgroup.shinyapps.io/aranoisy/). These data could also be used for other purposes, such as inferring regulatory networks based on gene expression correlation between seedlings, as previously done using microarrays of individual leaves (Bhosale *et al*, 2013).

We found that HVGs tend to be enriched for GOs involved in the response to environment, such as photosystems I and II, response to pathogens, response to abiotic stresses and response to iron ion. We also observed a high number of stress-responsive TFs with targets enriched in HVGs. This is in agreement with previous observations in mammals and yeast that HVGs are enriched in stress-responsive genes (Newman *et al*, 2006; Yin *et al*, 2009; Gasch *et al*, 2017) and that LVGs are enriched in housekeeping genes (Barroso *et al*, 2018). This is also further supported by previous results showing a positive correlation between gene expression variability and plasticity (Hirao *et al*, 2015), the latter corresponding to environmentally triggered gene expression changes. It was also proposed in single-celled organisms that transcriptional noise could be beneficial under unpredictable conditions (Kussell & Leibler, 2005; Freed *et al*, 2008; Zhuravel *et al*, 2010; Kellogg & Tay, 2015; Liu *et al*, 2015), a concept also known as bet hedging. In particular, gene expression variability for stress-responsive genes between cells in a population was associated with survival of a fraction of cells during stress treatment and reconstitution of the full population once favourable conditions returned (Levy *et al*, 2012; Grimbergen *et al*, 2015). It is interesting to note that we have found functional classes of highly variable genes that are similar to the ones found for variable genes in single-celled organisms. This is the case even though our work is at the whole plant scale, averaged over 10,000s of cells, which suggests similar but different mechanisms for the generation of this transcriptional variability. It would be of interest to define whether similar types of gene regulatory circuits are involved in the generation of this transcriptional variability and whether it could also originate from

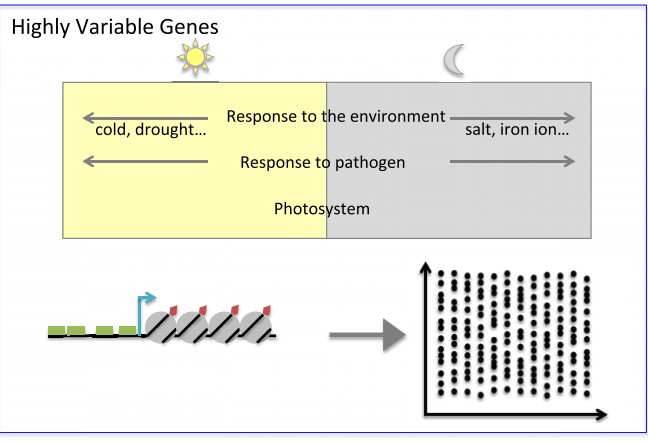

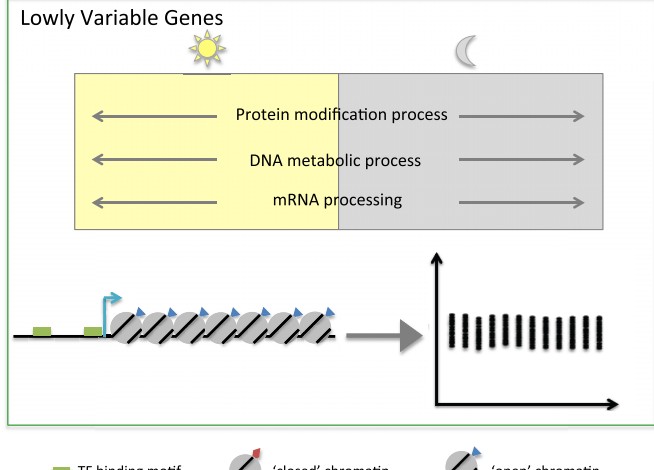

**Figure 6.  Models of HVGs and LVGs.**

HVGs (top panel) are enriched in environmentally responsive genes, with processes often more variable either during the day (yellow rectangle) or the night (grey rectangle). These genes tend to be smaller, to be targeted by a higher number of TFs (represented by the TFs binding motifs in green) and to have a more compacted chromatin environment (nucleosomes with red rectangles). On the other hand, LVGs (bottom panel) are enriched in genes involved in primary metabolism. These genes tend to be longer, to be targeted by a smaller number of TFs (represented by the TFs binding motifs in green) and to have a more open chromatin environment (nucleosomes with blue triangles), compared to HVGs.

variability in the stress level of seedlings or their responsiveness to the environment. Moreover, we do not know whether HVGs exhibit similar behaviour in different parts of the plant or whether these genes are more or less variable in different parts of the plant. Further analysis of inter-individual gene expression variability in different tissues (e.g. in roots, hypocotyls, aerial parts, etc.) would be required to answer this question. Given the high number of environmentally responsive genes among HVGs, it would be of interest to test whether inter-individual variability in stress-responsive genes could be correlated with variation in stress survival in *Arabidopsis thaliana*. This hypothesis is probable, as phenotypic variability has been observed for many traits in *Arabidopsis thaliana* (Paxman, 1956; Sakai & Shimamoto, 1965; Hall *et al*, 2007; Forde, 2009; Jimenez-Gomez *et al*, 2011). Moreover, the proportion of wild-type plants surviving a stress is not zero in many studies (Dai *et al*, 2007; Fasano *et al*, 2014;

Pitzschke *et al*, 2014; Silva-Correia *et al*, 2014), suggesting the possibility of underlying gene expression variability for stress-responsive genes explaining this observation. Our analysis of inter-individual gene expression variability was performed under non-stressed-controlled conditions, but in the future, it would be interesting to investigate how gene expression variability is influenced by changes in the environment and stress. Indeed, it was shown in yeast that genes coding for ribosomal proteins display a low level of variability in the absence of stress but become more variable during stress treatment (Gasch *et al*, 2017). On the other hand, genes involved in environmental stress response are highly variable in the absence of stress but show a reduction in their variability during stress treatment (Gasch *et al*, 2017).

We identified several genomic and epigenomic factors that are correlated with gene expression variability. We found that HVGs tend to be shorter and targeted by a higher number of TFs than LVGs. In line with our results, a negative correlation was also previously observed in yeast between gene length and noise for genes with a low plasticity (Bajić & Poyatos, 2012). It has also been shown in *Arabidopsis thaliana* that stress-responsive genes are shorter (Aceituno *et al*, 2008), in agreement with the fact that HVGs are enriched in environmentally responsive genes. Our results are further supported by previous studies showing that genetic factors can control or buffer inter-individual phenotypic variability in *Arabidopsis thaliana* (Hall *et al*, 2007; Jimenez-Gomez *et al*, 2011; Folta *et al*, 2014; Hong *et al*, 2016; Schaefer *et al*, 2017). Cis factors have also been shown to regulate gene expression variability in other organisms: TATA boxes are linked to gene expression variability in yeast (Blake *et al*, 2006), and the strength of cis-regulatory elements affects transcriptional noise in mammals (Suter *et al*, 2011). The diurnal profiles we observed in inter-individual variability however indicate that genetic factors can only make genes prone to be variable but are not sufficient to explain their variability level at a given time of the day. This indicates that other factors, such as gene regulatory networks for example, are involved in modulating the level of variability of a gene.

On top of genomic factors, we identified that HVGs and LVGs have distinct chromatin profiles, with HVGs being characterised by an enrichment in H3K27me1 and H3K27me3, which are repressive marks, and depleted in active marks such as H3K4me2, H3K4me3, H3K36me3, H2Bub or DNA methylation. The ChIP-seq data we used were obtained from bulk plant experiments, but in the future, it would be of interest to directly compare chromatin marks and expression levels by performing RNA-seq and ChIP-seq or BS-seq on the same individual seedling or cell. Although very challenging, recent advances on single-cell RNA-seq, ChIP-seq and BS-seq indicate that such types of experiment could be possible. Variability in DNA methylation was for example recently reported using single-cell approaches in human (Ecker *et al*, 2017; Garg *et al*, 2018). Chromatin has been shown to regulate the level of transcriptional noise, sometimes independently of expression level, in mammals (Wu *et al*, 2017; Barroso *et al*, 2018) and yeast (Weinberger *et al*, 2012). In plants, over-expression of CHR23, a chromatin remodeller, is associated with an increase in inter-individual phenotypic and transcriptional variability (Folta *et al*, 2014). These previous observations are in agreement with our results and suggest a role of the chromatin structure in regulating the level of gene expression variability, potentially with more compacted chromatin environments being more favourable to high variability. We nonetheless have to keep in mind that all these genomic and epigenomic factors are linked, as environmentally responsive genes have been shown to be smaller and to have a high gene body H2A.Z signal (Coleman-Derr & Zilberman, 2012), and that H3K27me3 was shown to be more enriched at small genes (Roudier *et al*, 2011). Our work has revealed the extent of gene expression variability between plants and how it might be regulated. It sets the stage for future work examining the potential function and specific mechanism of variability for each noisy pathway revealed here.

# Materials and Methods

**Reagents and Tools table**

| Reagent/resource | Reference or source | Identifier or catalog number |
|---|---|---|
| **Reagents and kits** | | |
| Solid 1/2X Murashige and Skoog media | Sigma-Aldrich | M5519-1L |
| MagMAX™-96 Total RNA Isolation Kit | Thermo Fisher | AM1830 |
| TruSeq Stranded mRNA Library Preparation Kit | Illumina | RS-122-2101 |
| ERCC RNA Spike-In Mix | Thermo Fisher | 4456740 |
| NextSeq 500/550 High Output v2 Kit (150 cycles) | Illumina | FC-404-2002 |
| Transcriptor First Strand cDNA Synthesis Kit | Sigma-Aldrich | 11483188001 |
| LC480 SYBR Green I Master | Roche | 04707516001 |
| **Software** | | |
| FastQC | www.bioinformatics.babraham.ac.uk/projects/fastqc/ | |
| TopHat | Trapnell *et al* (2009) | |
| Hisat2 | Kim *et al* (2015) | |
| Salmon | Patro *et al* (2017) | |

**Reagents and Tools table** (continued)

| Reagent/resource | Reference or source | Identifier or catalog number |
|---|---|---|
| Trimmomatic | Bolger *et al* (2014) | |
| R | https://www.r-project.org/ | |
| Bowtie2 | Langmead and Salzberg (2012) | |
| DeepTools | Ramirez *et al* (2014, 2016) | |

## Methods and Protocols

### Plant materials and growth conditions

Col-0 WT *Arabidopsis thaliana* seeds were sterilised, stratified for 3 days at 4°C in dark and transferred for germination on solid 1/2X Murashige and Skoog (MS) media at 22°C in long days for 24 h. Using a binocular microscope, seeds that were at the same stage of germination were transferred into a new plate containing solid 1/2X MS media. In total, 16 seeds were transferred into each of the 12 individual plates. Seedlings were grown at 22°C, 65% humidity, with 12 h of light (170 μmoles) and 12 h of dark in a conviron reach in cabinet. After 7 days of growth, seedlings were harvested individually into a 96-well plate and flash-frozen in dry ice (see Appendix Fig S1A for a photograph of seedlings grown in exactly the same conditions). Sixteen seedlings were harvested at each time-point, every 2 h over a 24-h period. In order to reduce environmental effects, all seedlings harvested for one time-point were growing in the same plate, and seedlings that looked smaller than others were not harvested. Moreover, the seedling number corresponds to the seedling position in the plate and we could not see any obvious position effect when analysing gene expression variability (Appendix Fig S2G). Only seedlings for which the root was on the surface of the MS media were harvested, in order to avoid breaking roots while harvesting. ZT2 to ZT12 corresponding to time-points harvested during the day, and ZT14 to ZT24 to time-points harvested during the night, and ZT12 and ZT24 being, respectively, harvested just a few minutes before dusk and dawn (Fig 1A). Night time-points were harvested in the dark using a green lamp in order to avoid any interruption of the dark period with white light.

### RNA-seq library preparation

Sixteen 7-day-old Col-0 WT *Arabidopsis* seedlings were harvested individually and flash-frozen in dry ice every 2 h over a 24-h period. Total RNA was isolated from 1 ground seedling using the MagMAX™-96 Total RNA Isolation Kit following manufacturer's recommendation. RNA quality and integrity were assessed on the Agilent 2200 TapeStation, and RNA concentration was assessed using Qubit RNA HS assay kit. Library preparation was performed using the TruSeq Stranded mRNA Library Preparation Kit (Illumina, RS-122-2101), for 1 μg of high-integrity total RNA (RIN > 8) into which 2 μl of diluted 1:100e ERCC RNA Spike-In Mix (Thermo Fisher, cat 4456740) was added. The libraries were sequenced on a NextSeq 500 using paired-end sequencing of 75 bp in length.

### RNA-seq mapping, identification of HVG and corrected CV² calculation

The raw reads were analysed using a combination of publicly available software and in-house scripts. We first assessed the quality of reads using FastQC (www.bioinformatics.babraham.ac.uk/projects/fastqc/). Potential adaptor contamination and low-quality trailing sequences were removed using Trimmomatic (Bolger *et al*, 2014), before aligned to the TAIR10 transcriptome using TopHat (Trapnell *et al*, 2009). Potential optical duplicates resulting from library preparation were removed using the Picard tools (https://github.com/broadinstitute/picard). For each gene, raw reads and TPM (transcripts per million) (Wagner *et al*, 2012) were computed. TPMs, which correct for gene length and library size, were used for all analyses.

Because of the high number of samples, RNA extraction and library preparation were performed in two batches, each batch containing half of the samples for each time-point. A batch effect was identified that can be explained by the plate in which the samples were for the library preparation. RUV function from the RUVseq R package (Risso *et al*, 2014) was used to remove this batch effect independently for each time-point.

Samples with at least 4 million reads were used, which is between 14 and 16 samples per time-point (Table EV1). To define the number of seedlings to use in order to identify transcriptional variability, we compared the corrected square coefficient of variation (corrected $CV^2$) obtained when analysing 6–15 seedlings with the ones obtained with 16 seedlings at the time-point ZT6, as we collected up to 16 seedlings for this time-point (Appendix Fig S1D). We observed a plateau in the increase in correlation from 10 or more seedlings, with a correlation of more than 0.9 between the corrected $CV^2$ calculated using 16 seedlings and the ones calculated with a least 12 seedlings (Appendix Fig S1D). As our data set contains 14–16 seedlings for each time-point, we thus decided to use 14 seedlings in all cases to be able to compare the time-points. When more than 14 seedlings were available for one time-point, we removed the extra seedlings with the lowest number of reads. This is higher than what was done in plants until now, as Folta and colleagues (Folta *et al*, 2014) analysed inter-individual expression variability for eight genes using six seedlings, and Brennecke and colleagues (Brennecke *et al*, 2013) analysed gene expression variability using scRNA-seq for seven cells.

Identification of HVGs was performed separately for each time-point as described previously (Brennecke *et al*, 2013), using the code from M3Drop R package (https://github.com/tallulandrews/M3Drop). Briefly, genes were first filtered so that (i) their averaged expression level between all 168 seedlings was of 5 TPM or more, (ii) they were at least 150 bp long, (iii) they had a TPM of 0 in < 5 seedlings for the analysed time-point, and (iv) their averaged expression level was of 5 TPM or more in the analysed time-point. Then, the fitted variance-mean dependence was calculated for each time-point (global $CV^2$ trend in Fig 1B) using the remaining genes, and genes for which the coefficient of variation significantly exceeds 10% with a FDR at 10% were selected as highly variable

(see Table EV2 for lists of HVGs and LVGs selected for each time-point).

For each gene, a corrected $CV^2$ was calculated in order to correct for the negative trend observed between $CV^2$ and the averaged expression level, as $\log_2(CV^2/\text{trend}$ for the same expression level).

Mean normalised gene expression was used when representing gene expression throughout the time course. It was calculated for each gene by dividing the expression level at a given time-point by the average expression across the entire time course for the same gene.

### RT–qPCR

Sixteen 7-day-old Col-0 WT *Arabidopsis thaliana* seedlings were harvested individually and flash-frozen in dry ice every 2 h over a 24-h period. Total RNA was isolated from 1 ground seedling. RNA concentration was assessed using Qubit RNA HS assay kit. cDNA synthesis was performed on 700 ng of DNAse-treated RNA using the Transcriptor First Strand cDNA Synthesis Kit. For RT-qPCR analysis, 0.4 μl of cDNA was used as template in a 10 μl reaction performed in the LightCycler 480 instrument using LC480 SYBR Green I Master. Gene expression relative to two control genes (SandF and PP2A) was measured (See Table EV7 for the list of primers used for RT–qPCR). Then, in order to directly compare RT–qPCR data with RNA-seq data, expression levels for each gene were normalised by the averaged expression level of that gene across all seedlings at all time-points.

### Hierarchical clustering

Hierarchical clustering was performed either on the $\log_2(CV^2/\text{trend})$ or on the mean normalised expression level using the statistical programme R (R Core Team 2014) using the function hclust on 1-Pearson correlation.

### Gene ontology enrichment analysis

To assess over-represented biological functions of the genes in different clusters, we performed the GO enrichment analysis using the Gene Ontology enrichment analysis. GO enrichment *P*-values were calculated using Gene Ontology Consortium enrichment analysis tool (Ashburner *et al*, 2000; Consortium, 2017). All the genes for which a corrected $CV^2$ was calculated were used as a background list for the enrichment analyses and this for each time-point separately.

### Other bioinformatics analyses of transcriptomic data

Shannon entropy from Roudier and colleagues (Roudier *et al*, 2011) was used to measure gene expression tissue specificity of HVGs, LVGs and the thousand sets of random genes. It was calculated using publicly available developmental expression series (Schmid *et al*, 2005), after filtering genes that showed no expression in any conditions.

TF-target analysis was done using the available data generated by DNA affinity purification coupled with sequencing (DAP-seq), which provides the list of *in vitro* targets for 529 TFs (O'Malley *et al*, 2016).

Gene length and number of introns were calculated using the TAIR10 annotation.

### ChIP-seq mapping and profiling

The lists of genes being marked by the analysed chromatin marks were obtained from Roudier and colleagues (Roudier *et al*, 2011)

and from Coleman-Derr and Zilberman (Coleman-Derr & Zilberman, 2012).

ChIP-seq data were downloaded from GSE101220 for H3K27me3 (Jiang & Berger, 2017), from GSE79355 for H2A.Z and MNase (Cortijo *et al*, 2017), from GSE73972 for H3K4me3 (Chen *et al*, 2017) and from GSE51304 for H3K23ac and H3 (Stroud *et al*, 2014). Sequenced ChIP-seq data were analysed in house, following the same quality control and pre-processing as in RNA-seq. The adaptor-trimmed reads were mapped to the TAIR10 reference genome using Bowtie2 (Langmead *et al*, 2009). Potential optical duplicates were removed using Picard, as described earlier. Averaged profiles and heatmap of the ChIP-seq signal along the gene body from 500 bp upstream to the transcription start site (TSS) to 500 bp downstream of the transcription termination site (TTS) were generated using deepTools (Ramirez *et al*, 2014, 2016). The ChIP signal was normalised by the INPUT, when available (for H3K27me3 and H3K4me3).

## Data availability

The data sets and computer code produced in this study are available in the following databases:

- RNA-seq data: Gene Expression Omnibus GSE115583: https://www.ncbi.nlm.nih.gov/geo/query/acc.cgi?acc = GSE115583.
- Graphical web interface: https://jlgroup.shinyapps.io/aranoisy/.
- Computer codes used to analyse RNA-seq data: https://github.com/scortijo/Scripts_noise_paper_MSB_2018.

**Expanded View** for this article is available online.

## Acknowledgements

We thank Dr Hugo Tavares (Sainsbury Laboratory Cambridge University, UK) for helping with the creation of the interactive web interface. We thank Dr Katie Abley (Sainsbury Laboratory Cambridge University, UK) and Dr Varodom Charoensawan (Mahidol University, Thailand) for critical reading of the manuscript draft. This research was made possible by a fellowship from the Gatsby Foundation (GAT3272/GLC). The work in the Locke laboratory is further supported by the European Research Council under the European Union's Seventh Framework Programme (FP/2007-2013)/ERC Grant Agreement 338060.

## Author contributions

SC conceived of the project. SC and JCWL designed the project. SC performed RNA-Seq experiments and analysed and interpreted the data. ZA performed the RT–qPCR for Fig 1C and Appendix Fig S1C and G. SA helped with the analysis of the DAP-seq data set for Figs 3C and 4C. SC and JCWL wrote the article, with SC writing the first draft.

## Conflict of interest

The authors declare that they have no conflict of interest.

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
