## [Review Process File · Molecular Systems Biology]

Widespread inter-individual gene expression variability in *Arabidopsis thaliana*

Sandra Cortijo, Zeynep Aydin, Sebastian Ahnert and James Locke.

Review timeline:

Submission date:	9 th August 2018
Editorial Decision:	18 th September 2018
Revision received:	3 rd December 2018
Editorial Decision:	6 th December 2018
Revision received:	7 th December 2018
Accepted:	11 th December 2018

Editor: Maria Polychronidou

Transaction Report:

1st Editorial Decision

18th September 2018

Thank you again for submitting your work to Molecular Systems Biology. We have now heard back from the three referees who agreed to evaluate your study. As you will see below, the reviewers are positive and think that the study represents a useful resource for the field. They raise however a series of concerns, which we would ask you to address in a revision.

I think that the recommendations of the reviewers are quite clear, so there is no need to repeat the points listed below. Please do not hesitate to contact me in case you would like to further discuss any of the issues raised by the reviewers. Regarding the comment of reviewer #2 referring to the lack of mechanistic insights, we do not think that delineating specific mechanisms is required for the acceptance of the study for publication. However, we would encourage you to include the gene regulatory network analysis suggested by reviewer #2, to further examine the potential contribution of TFs to the observed variation.

REFeree REPORTS

Reviewer #1:

This manuscript explores variation in gene expression among individual *Arabidopsis* plants. This is an interesting topic because there is a growing awareness that even among genetically identical individuals there can be considerable variation in gene expression. Ultimately this may lead to a better understanding of stochasticity in phenotype and in environmental effects on development, disease, and morbidity. While this topic has received significant attention in microbes, there are few studies in plants and this explores the characteristics of genes with high or low variability in expression in far more detail than other studies that I am aware of. While mostly descriptive and correlative, this work serves as an excellent foundation for future studies by its in depth analyses. Writing is very clear, conclusions are justified.

Major points:

1. The authors define a set of highly variable genes (HVG) and then compare this to a set of random genes of similar size. The general standard when this type of empirical control is used is to create many random gene sets (100 or 1000) and then compare the test set (HVG in this case) to the distributional properties of the random sets. This allows statistical conclusions to be drawn about the differences between the test set (HVG) and random expectation. Shouldn't this approach be taken here?

2. Some important references are missing. The statement "gene expression variability has only been analysed for a few individual genes in plants" is incorrect. For example, Dan Kliebenstein's lab has also explored variation in Arabidopsis gene expression. The correct paper is actually cited (Jimenez-Gomez et al, 2011) but only in the context of phenotypic variance, not genome-wide expression variation. The prior work should be properly acknowledged and current results compared to previous findings. The Jimenez-Gomez paper has a very different focus than this manuscript so the prior work does not diminish the impact of the current manuscript. Similarly, Lin et al (G3, 2016) have explored these questions in Drosophila. This work should also be discussed and results compared to the current findings. I did not do an exhaustive literature search, but since these papers were missed I would encourage the authors to explore the literature and make sure there are not papers beyond these two that are relevant.

Minor points:

line 181: typo "is higher that between"

Fig 2C, S3C, S3D. The same cutoffs for heatmap shading should be used in the figures so that it is easy to compare. One approach would be to base this on p-value (or $-\log_{10}(p)$) for significant overlap based on Fisher's exact test.

Fig S4. Need information on how tissue specificity was ascertained. What data set? What analysis?

line 277: The 0.4 and -0.4 cutoffs seem rather arbitrary. Justify. Or better, use a significance cutoff instead.

Figure 4: x-axis labels should match what is being used in the text: HVG, LVG, random.

lines 290-295. A statistical approach to the gene length vs # of introns question would be useful. which has a larger R^2 in linear regression? If you start with intron# as the explanatory variable is a better fit obtained when gene length is added? how about the other way around?

Figure S1 legend. "pearson" should be capitalized (multiple occurrences).

Reviewer #2:

The manuscript by Cortijo and colleagues describes a transcriptomics resource containing 168 datasets derived from profiling 14 individual Arabidopsis seedlings at 12 time points over the course of one day. The manuscript delves into the analysis of gene expression variation and a number of interesting observations are presented. Highly variable genes of several classes are identified and their function and regulation are analysed. Some of the take home messages are that variable genes tend to be associated with functions in environmental response and that these genes are on average shorter and embedded in more repressive chromatin environments compared to random genes.

The data and analysis presented is very solid and represents a beautiful resource for the community, however it falls a bit short of my expectation when it comes to elucidating the mechanisms behind the observed gene expression variation. The authors identify 60 transcription factors enriched upstream of highly variable genes, but do not analyse their potential contribution to variation. The

claim that it is unlikely that variation is encoded at the TF level, because only seven of them are variable themselves, is weak, since a single variable input into a highly connected network with many nodes can result in global variation. The authors should therefore attempt to reconstruct a gene regulatory network from the HVGs and their transcription factors to analyse this aspect more rigorously.

Minor points: The manuscript is somewhat difficult to read here and there since the authors like to use the term "detected" in a number of contexts. To me, in the context of gene expression profiling "detected" stands for evidence of expression more than anything else.

Reviewer #3:

In this manuscript Cortijo and colleagues describe a new transcriptomics resource for *A. thaliana*. The authors have generated RNA-seq profiles for 168 plant seedlings at 12 time points during a 24h period. The originality of their approach is the high number of replicates (14) analysed at each time-points. This permitted to calculate a variability score for each gene at each time-points providing insight into the levels and the dynamics of gene expression variability during the day. Importantly this resource is made available to the wider community through a web interface. To showcase their dataset the authors defined HVGs at each time-points and found that these were involved preferentially in response to external conditions, were short, had many TF binding-sites and had a repressive chromatin architecture (based on published data).

This is a well written paper describing a great resource. The web interface set up by the authors is a very significant strength of this work which will ensure easy access by the community. I have a few comments and suggestions, which I hope will help improve the manuscript.

Major:

1) I am not familiar with *A. thaliana* biology and life cycle as it may be the case of many readers. In order to provide an accurate description their biological system, I think the authors should include on figure 1 a real picture of plants grown in conditions identical to those used in the paper. This would help the reader to understand the extent of phenotypic diversity present in the samples and to get a feeling about how similar each plant environment is on the plate.

2) The authors talk about gene expression variability between seedlings. This variability is apparent after averaging expression levels over thousands of cells as each RNA-seq library was made from a single whole organism. It would be useful if the authors could elaborate a bit in their introduction about what this form of variability really is. It clearly is not the sort of noise observed between single prokaryotic cells for instance. Related to the point above, do the authors assume that the 14 seedlings were grown in identical environments? For instance are seedling growing at the edges of the plate different than those from the middle?

3) Moreover, how many cell types are there in these seedlings? Could the relative proportion of different cell types participate to the observed variability?

4) On Figure 3E, the mean normalised expression ranges from 0.8 to 1.2, while on Figure 1B the average read counts span two orders of magnitude. I have not done the maths but this seems strange. Why have the authors chosen to compress the colour bar so much on Figure 3E? Aren't we missing some information here?

5) On Figure S5B, I really think the authors have to investigate the bimodal distribution of HVGs correlations with expression levels further. By this I mean consider the genes from each two peaks as a different group. Are those with high CV-mean correlations specifically induced during the day or the night for instance? Are those from the peak with low correlation lower or higher expressed? Etc...

6) Regarding the analysis of histone modifications. I think the authors should mention at the beginning of the paragraph how their ChIP-seq data compare to their time course in term of

experimental conditions. Where they acquired during the day? Or during the night?

7) Moreover, do the authors think that HVGs are expressed despite their repressive context? Or rather, that their chromatin structure is also variable (which would have been averaged out in the published ChIP-seq data)?

Minor:

1) Figure 3C, please provide the p-values for the red and green dots.

2) Authors find that HVGs tend to be short. Are short genes associated with any specific GO category in *A. thaliana*? If stress response genes were to be, it could explain this enrichment.

3) Line 405, "similar but different" mechanisms of generation of transcriptional variability. If I get this correctly, regulation at the level of one genome in individual cells is compared to a form of coordinate response of many cells at the organism level. What is "similar"? Do the authors think that some seedling are more responsive than others? Or rather that some are experiencing higher levels of stress?

We thank the reviewers for their positive appreciation of the manuscript and very useful comments and suggestions. We have included results and corrections as recommended by the reviewers, which we believe have significantly improved the manuscript. A detailed response to all comments can be found below.

Reviewer #1:

This manuscript explores variation in gene expression among individual Arabidopsis plants. This is an interesting topic because there is a growing awareness that even among genetically identical individuals there can be considerable variation in gene expression. Ultimately this may lead to a better understanding of stochasticity in phenotype and in environmental effects on development, disease, and morbidity. While this topic has received significant attention in microbes, there are few studies in plants and this explores the characteristics of genes with high or low variability in expression in far more detail than other studies that I am aware of. While mostly descriptive and correlative, this work serves as an excellent foundation for future studies by its in depth analyses. Writing is very clear, conclusions are justified.

Major points:

1. The authors define a set of highly variable genes (HVG) and then compare this to a set of random genes of similar size. The general standard when this type of empirical control is used is to create many random gene sets (100 or 1000) and then compare the test set (HVG in this case) to the distributional properties of the random sets. This allows statistical conclusions to be drawn about the differences between the test set (HVG) and random expectation. Shouldn't this approach be taken here?

Thank you for this suggestion. We have now created 1000 sets of random genes of the same size of the HVGs to compare with HVGs for the distributions of the corrected CV^2 , the number of time-points where genes are selected, gene length, number of introns, number of TFs targeting each gene, gene expression tissue specificity (entropy), gene expression level and for the correlation between profiles in gene expression and variability. The average and 95% interval calculated from these 1000 sets are now used in the corresponding figures.

2. Some important references are missing. The statement "gene expression variability has only been analysed for a few individual genes in plants" is incorrect. For example, Dan Kliebenstein's lab has also explored variation in Arabidopsis gene expression. The correct paper is actually cited (Jimenez-Gomez et al, 2011) but only in the context of phenotypic variance, not genome-wide expression variation. The prior work should be properly acknowledged and current results compared to previous findings. The Jimenez-Gomez paper has a very different focus than this manuscript so the prior work does not diminish the impact of the current manuscript. Similarly, Lin et al (G3, 2016) have explored these questions in Drosophila. This work should also be discussed and results compared to the current findings. I did not do an exhaustive literature search, but since these papers were missed I would encourage the authors to explore the literature and make sure there are not papers beyond these two that are relevant.

Thanks for this comment. We have now included these references (and others) in the

introduction and also described more the meaning of inter-individual transcriptional variability.

Minor points:

line 181: typo "is higher that between"

Corrected.

Fig 2C, S3C, S3D. The same cutoffs for heatmap shading should be used in the figures so that it is easy to compare. One approach would be to base this on p-value (or $-\log_{10}(p)$) for significant overlap based on Fisher's exact test.

Thanks for this suggestion. We have now added Appendix Fig S3G that includes heatmaps for HVGs, LVGs and one set of random genes using the same cutoffs of the heatmap shading.

Fig S4. Need information on how tissue specificity was ascertained. What data set? What analysis?

Thanks for this comment. We have added the following information in the material and methods section:

L623 “Shannon entropy from Roudier and colleagues (Roudier et al., 2011) was used to measure gene expression tissue specificity of HVGs, LVGs and the thousand sets of random genes. It was calculated using publicly available developmental expression series (Schmid et al., 2005), after filtering genes that showed no expression in any conditions.”

line 277: The 0.4 and -.4 cutoffs seem rather arbitrary. Justify. Or better, use a significance cutoff instead.

Thanks for this suggestion. We now use a significance cutoff of p-value less or equal to 0.05 and find 285 HVGs with a significant correlation between profiles of expression levels and variability (20% of all 1358 HVGs). We changed this in the manuscript.

L304: “If we consider HVGs with a significant correlation (p-value less or equal to 0.05), it seems that profiles in gene expression variability for approximately 20% of HVGs could be potentially explained by expression profiles (for profiles of positive and negative correlations see examples in Appendix Fig S5C-D).”

Figure 4: x-axis labels should match what is being used in the text: HVG, LVG, random.

Figure 4 was changed accordingly to major point 1, which also solves this point.

lines 290-295. A statistical approach to the gene length vs # of introns question would be useful. which has a larger R^2 in linear regression? If you start with intron# as the explanatory variable is a better fit obtained when gene length is added? how about the other way around?

Thanks for this suggestion. We have now performed this analysis and unfortunately a linear regression is not the best way to capture the relation between the $\log_2(CV^2/\text{trend})$ and the gene length or number of introns, as you can see in the figures below. This can also explain the low R^2 values in the table below. We thus decided to not include these results in the manuscript.

Time point	R2 of the linear fit		R2 of the linear fit	
	CorCV2~number of intron	CorCV2~gene length	CorCV2~number of intron + gene length	
ZT2	0.036	0.108	0.109	
ZT4	0.021	0.060	0.060	
ZT6	0.014	0.048	0.049	
ZT8	0.001	0.004	0.005	
ZT10	0.004	0.013	0.013	
ZT12	0.015	0.045	0.046	
ZT14	0.017	0.058	0.059	
ZT16	0.019	0.062	0.064	
ZT18	0.028	0.111	0.116	
ZT20	0.028	0.091	0.093	
ZT22	0.041	0.136	0.139	
ZT24	0.022	0.085	0.088	

Figure S1 legend. "pearson" should be capitalized (multiple occurrences).
Thanks for this remark; this has been changed throughout the manuscript.

Reviewer #2:

The manuscript by Cortijo and colleagues describes a transcriptomics resource containing 168 datasets derived from profiling 14 individual Arabidopsis seedlings at 12 time points over the course of one day. The manuscript delves into the analysis of gene expression variation and a number of interesting observations are presented. Highly variable genes of several classes are identified and their function and regulation are analysed. Some of the take home messages are that variable genes tend to be associated with functions in environmental response and that these genes are on average shorter and embedded in more repressive chromatin environments compared to random genes.

The data and analysis presented is very solid and represents a beautiful resource for the community, however it falls a bit short of my expectation when it comes to elucidating the mechanisms behind the observed gene expression variation. The authors identify 60 transcription factors enriched upstream of highly variable genes, but do not analyse their potential contribution to variation. The claim that it is unlikely that variation is encoded at the TF level, because only seven of them are variable themselves, is weak, since a single variable input into a highly connected network with many nodes can result in global variation. The authors should therefore attempt to reconstruct a gene regulatory network from the HVGs and their transcription factors to analyse this aspect more rigorously.

Thanks for this suggestion. We have now analysed more in detail the regulation of HVGs by the 60 TFs with targets enriched in HVGs, of which 7 TFs are themselves highly variable. As suggested, we also derived gene regulatory networks for HVGs and these TFs based on the DAP-seq data. We added the following in the results: L251: "1106 out of 1358 HVGs are potential targets of at least one of these 7 TFs. However 23301 genes in total are potential targets of at least one of these 7 TFs, so only a small fraction of these potential targets are HVGs (Table EV5). Moreover, DAP-seq data being derived from in vitro interaction provides a list of potential targets and further experiments such as ChIP-seq would be required to obtain the list of genes regulated by these TFs in our conditions. When deriving gene regulatory networks from the DAP-seq data for HVGs and these TFs, we observe a high level of regulation of these 60 TFs by other TFs of this same list, and that most HVGs are targeted by a combination of highly variable and non-highly variable TFs (Appendix Fig S4C-E and Table EV6). These results suggest that while the high level of variability could potentially partly be explained by TFs, other factors are also probably involved."

However, we have to keep in mind that DAP-seq is performed in vitro and is thus only providing a list of potential targets. We do not know what is the proportion of these potential targets that are actually regulated by the TFs. Other experiments, such as ChIP-seq and RNA-seq of mutants for these TFs, would be required to refine these gene regulatory networks.

We would also like to add that we are currently performing a network analysis of the transcriptomic dataset of this manuscript, which will be the main focus of a paper under preparation.

Minor points: The manuscript is somewhat difficult to read here and there since the authors like to use the term "detected" in a number of contexts. To me, in the context of gene expression profiling "detected" stands for evidence of expression more than anything else.

Thanks for this remark; this has been changed throughout the manuscript.

Reviewer #3:

In this manuscript Cortijo and colleagues describe a new transcriptomics resource for *A. thaliana*. The authors have generated RNA-seq profiles for 168 plant seedlings at 12 time points during a 24h period. The originality of their approach is the high number of replicates (14) analysed at each time-points. This permitted to calculate a variability score for each gene at each time-points providing insight into the levels and the dynamics of gene expression variability during the day. Importantly this resource is made available to the wider community through a web interface. To showcase their dataset the authors defined HVGs at each time-points and found that these were involved preferentially in response to external conditions, were short, had many TF binding-sites and had a repressive chromatin architecture (based on published data).

This is a well written paper describing a great resource. The web interface set up by the authors is a very significant strength of this work which will ensures easy access by the community. I have a few comments and suggestions, which I hope will help improve the manuscript.

Major:

1) I am not familiar with *A. thaliana* biology and life cycle as it may be the case of many readers. In order to provide an accurate description their biological system, I think the authors should include on figure 1 a real picture of plants grown in conditions identical to those used in the paper. This would help the reader to understand the extent of phenotypic diversity present in the samples and to get a feeling about how similar each plant environment is on the plate.

Thanks for this suggestion. We grew plants under identical conditions to those used in our experiments and took a picture that is now in Appendix Fig S1A.

2) The authors talk about gene expression variability between seedlings. This variability is apparent after averaging expression levels over thousands of cells as each RNA-seq library was made from a single whole organism. It would be useful if the authors could elaborate a bit in their introduction about what this form of variability really is. It clearly is not the sort of noise observed between single prokaryotic cells for instance. Related to the point above, do the authors assume that the 14 seedlings were grown in identical environments? For instance are seedling growing at the edges of the plate different than those from the middle?

Thanks for this comment. We have now included the following sentences in the introduction and discussion of the manuscript to answer this point:

L55: “It is not known if such inter-individual phenotypic variability originates from responses to microenvironmental perturbations or from stochastic factors at the cellular level, or from both.”

L445: “Moreover, we do not know if HVGs exhibit similar behaviour in different parts of the plant, or whether these genes are more or less variable in different parts of the plant. Further analysis of inter-individual gene expression variability in different tissues (e.g in roots, hypocotyls, aerial parts etc...) would be required to answer this question.”

Moreover precautions were taken during the experiment to reduce developmental and environmental variability. As explained in the material and methods, we could not see a plate effect when analysing gene expression.

L524: “In order to reduce environment effects, all seedlings harvested for one time-point were growing in the same plate, and seedlings that looked smaller than others were not harvested. Moreover, the seedling number corresponds to the seedling position in the plate and we could not see any obvious position effect when analysing gene expression variability (Appendix Fig S2G). Only seedlings for which the root was on the surface of the MS media were harvested, in order to avoid breaking roots while harvesting.”

We now also included heatmaps of the hierarchical clustering, for each time-point, of HVGs and of individual seedling using the mean normalised expression levels, further showing that seedlings at the edged of the plate do not show similar expression profiles (Appendix Fig S2G).

3) Moreover, how many cell types are there in these seedlings? Could the relative proportion of different cell types participate to the observed variability?

The first point is a very interesting question to which there is no consensus. We can only say that an Arabidopsis seedling contains a high number of different cell types. About the second point, we now analysed the expression level of the HVGs in the different tissues using the same dataset we used for the entropy analysis. We find that most HVGs are expressed in more than one tissue suggesting that the relative proportion of different cell types would not be the primary cause of the observed variability (Appendix Fig S4B).

4) On Figure 3E, the mean normalised expression ranges from 0.8 to 1.2, while on Figure 1B the average read counts span two orders of magnitude. I have not done the maths but this seems strange. Why have the authors chosen to compress the colour bar so much on Figure 3E? Aren't we missing some information here?

Thanks for this remark. On Fig 3E gene expression for each gene is normalised by its average expression level across the time-course. This is not normalised by the average of all genes in one time-point and thus cannot be directly compared with Fig 1B. We have added more information on the material and section to avoid any confusion.

L591: “Mean normalised gene expression was used when representing gene expression throughout the time course. It was calculated for each gene by dividing the expression level at a given time-point by the average expression across the entire time course for the same gene.”

5) On Figure S5B, I really think the authors have to investigate the bimodal distribution of HVGs correlations with expression levels further. By this I mean

consider the genes from each two peaks as a different group. Are those with high CV-mean correlations specifically induced during the day or the night for instance? Are those from the peak with low correlation lower or higher expressed? Etc...

Thanks for this comment. We have now explored the HVGs under the two peaks and compared their profiles in expression, their expression levels and the number of time points when they are identified as highly variable. We could only find a difference for the expression level, for which HVGs with a positive correlation between variability and expression levels have a lower expression level in general compared to other HVGs. We have integrated these results in the manuscript:

L300: “We cannot see major differences in the expression profiles or the number of time-points for which genes are identified as HVGs for these two groups of genes (Appendix Fig S5C-D). However, HVGs with a positive correlation between variability and expression levels (peak around 0.5 in Appendix Fig S5B) have a lower expression level in general compared to other HVGs (Appendix Fig S5E).”

6) Regarding the analysis of histone modifications. I think the authors should mention at the beginning of the paragraph how their ChIP-seq data compare to their time course in term of experimental conditions. Where they acquired during the day? Or during the night?

Thanks for this question. ChIP-seq data are extracted from available datasets and we unfortunately don't have information about the time of the day at which the plants were harvested to perform the ChIP-seq. The following has been added in the manuscript to clarify this point:

L356: “we analysed several histone marks using data already available for which we have no information about the time of day when the plants where harvested.”

7) Moreover, do the authors think that HVGs are expressed despite their repressive context? Or rather, that their chromatin structure is also variable (which would have been averaged out in the published ChIP-seq data)?

This is an interesting point. Correlation between expression variability and chromatin state has also been observed in previous studies in other organism and we do not know yet if, in one cell, a gene is expressed while despite a repressive environment. We added the following in the discussion to reflect this point:

L489: “The ChIP-seq data we used were obtained from bulk plant experiments, but in the future it would be of interest to directly compare chromatin marks and expression levels by performing RNA-seq and ChIP-seq or BS-seq on the same individual seedling or cell. Although very challenging, recent advances on single-cell RNA-seq, ChIP-seq and BS-seq indicate that such types of experiment could be possible. Variability in DNA methylation was for example recently reported using single-cell approaches in human (Ecker et al., 2017; Garg et al., 2018).”

Minor:

1) Figure 3C, please provide the p-values for the red and green dots.

Thanks for this suggestion. p-values of the Fisher's exact test have been added to the Fig 3C.

2) Authors find that HVGs tend to be short. Are short genes associated with any specific GO category in *A. thaliana*? If stress response genes were to be, it could explain this enrichment.

Thanks. This is indeed the case, and this point is included in the discussion of the paper:

L471: “It has also been shown in *Arabidopsis thaliana* that stress responsive genes are shorter (Aceituno et al., 2008), in agreement with the fact that HVGs are enriched in environmentally responsive genes.”

3) Line 405, "similar but different" mechanisms of generation of transcriptional variability. If I get this correctly, regulation at the level of one genome in individual cells is compared to a form of coordinate response of many cells at the organism level. What is "similar"? Do the authors think that some seedling are more responsive than others? Or rather that some are experiencing higher levels of stress?

Thanks for this comment. The following sentence was included in the discussion of the manuscript to take into account this point:

L442: “It would be of interest to define if similar types of gene regulatory circuits are involved in the generation of this transcriptional variability, and whether it could also originate from variability in the stress level of seedlings or their responsiveness to the environment.”

Thank you for sending us your revised manuscript. We are satisfied with the modifications made and we think that the study is now suitable for publication.

Before we formally accept the study for publication, we would ask you to address the following minor editorial issues:

Corresponding Author Name: James Locke
Journal Submitted to: Molecular Systems Biology
Manuscript Number: MSB-18-8591